# Three-dimensional electron crystallography of protein microcrystals

Dan Shi[†], Brent L Nannenga[†], Matthew G Iadanza[†], Tamir Gonen*

Janelia Farm Research Campus, Howard Hughes Medical Institute, Ashburn, United States

**Abstract** We demonstrate that it is feasible to determine high-resolution protein structures by electron crystallography of three-dimensional crystals in an electron cryo-microscope (CryoEM). Lysozyme microcrystals were frozen on an electron microscopy grid, and electron diffraction data collected to 1.7 Å resolution. We developed a data collection protocol to collect a full-tilt series in electron diffraction to atomic resolution. A single tilt series contains up to 90 individual diffraction patterns collected from a single crystal with tilt angle increment of 0.1–1° and a total accumulated electron dose less than 10 electrons per angstrom squared. We indexed the data from three crystals and used them for structure determination of lysozyme by molecular replacement followed by crystallographic refinement to 2.9 Å resolution. This proof of principle paves the way for the implementation of a new technique, which we name 'MicroED', that may have wide applicability in structural biology.

## Introduction

X-ray crystallography depends on large and well-ordered crystals for diffraction studies. Crystals are solids composed of repeated structural motifs in a three-dimensional lattice (hereafter called '3D crystals'). The periodic structure of the crystalline solid acts as a diffraction grating to scatter the X-rays. For every elastic scattering event that contributes to a diffraction pattern there are ~10 inelastic events that cause beam damage (*Henderson, 1995*). Therefore, large crystals are required to withstand the high levels of radiation damage received during data collection (*Henderson, 1995*). Despite the development of highly sophisticated robotics for crystal growth assays and the implementation of microfocus beamlines (*Moukhametzianov et al., 2008*), this important step remains a critical bottleneck. In an attempt to alleviate this problem, researchers have turned to femtosecond X-ray crystallography (*Chapman et al., 2011*; *Boutet et al., 2012*), in which a very intense pulse of X-rays yields coherent signal in a time shorter than the destructive response to deposited energy. While this technique shows great promise, the current implementation of the technology requires an extremely large number of crystals (millions) and access to sources is still in developmental stages.

Electron crystallography is a bona fide method for determining protein structure from crystalline material but with important differences. The crystals that are used must be very thin (*Henderson and Unwin, 1975*; *Henderson et al., 1990*; *Kuhlbrandt et al., 1994*; *Kimura et al., 1997*). Because electrons interact with materials more strongly than X-rays (*Henderson, 1995*), electrons can yield meaningful data from relatively small and thin crystals. This technique has been used successfully to determine the structures of several proteins from thin two-dimensional crystals (2D crystals) (*Wisedchaisri et al., 2011*). High energy electrons result in a large amount of radiation damage to the sample, leading to loss in resolution and destruction of the crystalline material (*Glaeser, 1971*). As each crystal can usually yield only a single diffraction pattern, structure determination is only possible by merging data originating from hundreds of individual crystals. For example, electron diffraction data from more than 200 individual crystals were merged to generate a data set for aquaporin-0 at 1.9 Å resolution (*Gonen et al., 2005*).

*For correspondence: gonent@ janelia.hhmi.org

†These authors contributed equally to this work

Competing interests: The authors declare that no competing interests exist.

**eLife digest** X-ray crystallography has been used to work out the atomic structure of a large number of proteins. In a typical X-ray crystallography experiment, a beam of X-rays is directed at a protein crystal, which scatters some of the X-ray photons to produce a diffraction pattern. The crystal is then rotated through a small angle and another diffraction pattern is recorded. Finally, after this process has been repeated enough times, it is possible to work backwards from the diffraction patterns to figure out the structure of the protein.

The crystals used for X-ray crystallography must be large to withstand the damage caused by repeated exposure to the X-ray beam. However, some proteins do not form crystals at all, and others only form small crystals. It is possible to overcome this problem by using extremely short pulses of X-rays, but this requires a very large number of small crystals and ultrashort X-ray pulses are only available at a handful of research centers around the world. There is, therefore, a need for other approaches that can determine the structure of proteins that only form small crystals.

Electron crystallography is similar to X-ray crystallography in that a protein crystal scatters a beam to produce a diffraction pattern. However, the interactions between the electrons in the beam and the crystal are much stronger than those between the X-ray photons and the crystal. This means that meaningful amounts of data can be collected from much smaller crystals. However, it is normally only possible to collect one diffraction pattern from each crystal because of beam induced damage. Researchers have developed methods to merge the diffraction patterns produced by hundreds of small crystals, but to date these techniques have only worked with very thin two-dimensional crystals that contain only one layer of the protein of interest.

Now Shi et al. report a new approach to electron crystallography that works with very small three-dimensional crystals. Called MicroED, this technique involves placing the crystal in a transmission electron cryo-microscope, which is a fairly standard piece of equipment in many laboratories. The normal 'low-dose' electron beam in one of these microscopes would normally damage the crystal after a single diffraction pattern had been collected. However, Shi et al. realized that it was possible to obtain diffraction patterns without severely damaging the crystal if they dramatically reduced the normal low-dose electron beam. By reducing the electron dose by a factor of 200, it was possible to collect up to 90 diffraction patterns from the same, very small, three-dimensional crystal, and then—similar to what happens in X-ray crystallography—work backwards to figure out the structure of the protein. Shi et al. demonstrated the feasibility of the MicroED approach by using it to determine the structure of lysozyme, which is widely used as a test protein in crystallography, with a resolution of 2.9 Å. This proof-of principle study paves the way for crystallographers to study protein that cannot be studied with existing techniques.

While electron crystallography has been successful with 2D crystals, previous attempts at using electron diffraction for structure determination from protein 3D crystals were not successful. A number of studies detail the difficulties associated with data collection and processing of diffraction data that originates from several hundreds of 3D crystals, limiting the ability to integrate and merge the data in order to determine a structure in such a way (*Shi et al., 1998*; *Jiang et al., 2011*).

We show here that atomic resolution diffraction data can be collected from crystals with volumes up to six orders of magnitude smaller than those typically used for X-ray crystallography. The technique, which we call 'MicroED', uses equipment standard in most cryo-EM laboratories and facilities. We developed a strategy for data collection with extremely low electron dose and procedures for indexing and integrating reflections. We processed the diffraction data and determined the structure of lysozyme at 2.9 Å resolution. Thus, a high-resolution protein structure can be determined from electron diffraction of three-dimensional protein crystals in an electron microscope.

## Results

### Sample preparation and data collection

Lysozyme was chosen as a model protein because it is a well-behaved and well-characterized protein that readily forms well-ordered crystals. From the time its structure was first analyzed (*Blake et al.,*

*1962*, *1965*), lysozyme has been a well-studied protein and the model protein of choice for many new methods in crystallography (*Boutet et al., 2012*; *Cipriani et al., 2012*; *Nederlof et al., 2013*). Small microcrystals of lysozyme were grown by slightly modifying the crystal growth conditions as detailed in the 'Materials and methods' section. *Figure 1A* shows a typical crystallization drop containing microcrystals, which appear as barely visible specks (arrows) alongside the larger crystals that are typically used for X-ray crystallography. These specks are up to 6 orders of magnitude smaller in volume than the larger crystals in the drop. The solution containing these microcrystals was applied to an electron microscopy holey-carbon grid with a pipette and plunged into liquid ethane. The grids were then imaged using a 200 kV TEM under cryogenic conditions (*Figure 1B*). More than 100 micro-crystals were typically observed per grid preparation, and these ranged in size from several microns to sub micron. The crystals typically appeared as electron dense rectangular or triangular forms with very sharp edges.

Electron diffraction was used to assess the quality of the cryo-preparations. Crystals that appeared thick (estimated as >3 μm) did not yield diffraction data because the electron beam could not penetrate the sample. Crystals that appeared slightly thinner, estimated at ~1.5 μm, did show diffraction, but because the quality of the pattern varied depending on the sample tilt (*Figure 2A*), we did not use crystals of this thickness and size for data collection. Approximately 50% of the crystals in our preparations appeared much thinner, estimated at ~0.5 μm, and showed a distribution of attainable resolutions with the best diffracting to ~1.7 Å resolution (*Figure 2B,C*). Generally, we were only able to obtain high quality diffraction data from the very thin crystals, ~0.5–1 μm thick and 1–6 μm long and wide. While these crystals are exceptionally small, they still contain approximately $55 \times 10^6$ unit cells. Moreover, we found that for such thin crystals the tilt had no significant adverse affect on the diffraction quality (*Figure 2D*).

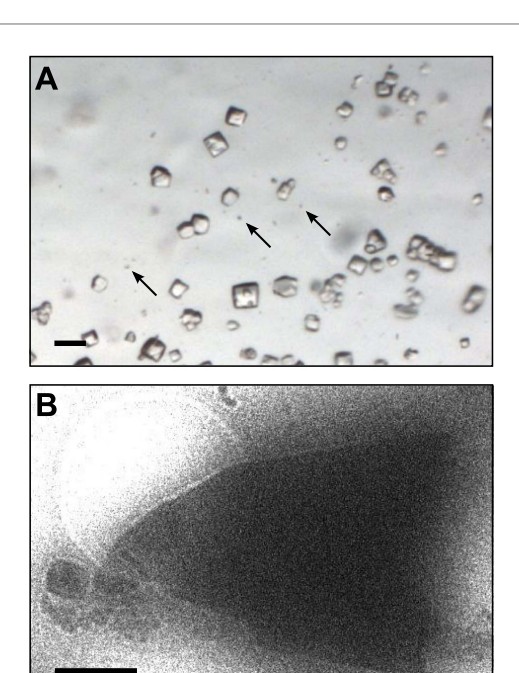

**Figure 1**. Images of lysozyme microrystals. (**A**) Light micrograph showing lysozyme microcrystals (three examples indicated by arrows) in comparison with larger crystals of the size normally used for X-ray crystallography. Scale bar is 50 μm. (**B**) Lysozyme microcrystals visualized in over-focused diffraction mode on the cryo-EM prior to data collection. The length and width of the crystals varied from 2 to 6 μm with an estimated thickness of ~0.5–1 μm. Scale bar is 1 μm.

For 2D electron crystallography, the electron dose that is typically used in diffraction causes significant radiation damage to the sample, leading to a rapid loss in resolution and destruction of the crystal (*Glaeser, 1971*; *Unwin and Henderson, 1975*; *Taylor and Glaeser, 1976*). As a result, each crystal exposed to high dose usually only yields a single diffraction pattern, and structure determination requires the merging of data originating from a large number of individual crystals. However, 3D crystals can deliver electron diffraction data to atomic resolution with very low doses. A recent study documents ~3 Å resolution diffraction data from catalase 3D crystals after a single exposure of less than $10e^-/Å^2$ (*Baker et al., 2010*).

We reasoned that one way to overcome the difficulties of indexing and merging data from hundreds of crystals is to collect a complete diffraction data set from a single crystal while keeping the total dose below ~$10e^-/Å^2$. Because all the data would originate from a single crystal, indexing, integration and merging should be straightforward and structure determination possible. We used a sensitive CMOS based detector (Tietz Video and Image Processing Systems GmbH), previously shown to be beneficial for electron diffraction studies (*Tani et al., 2009*) and modified our data collection procedure. We found that even with extremely low electron dose of <0.01 $e^-/Å^2$ per second, we could record diffraction data from lysozyme microcrystals showing strong and sharp diffraction spots extending well beyond the 2 Å resolution mark (*Figure 2C*).

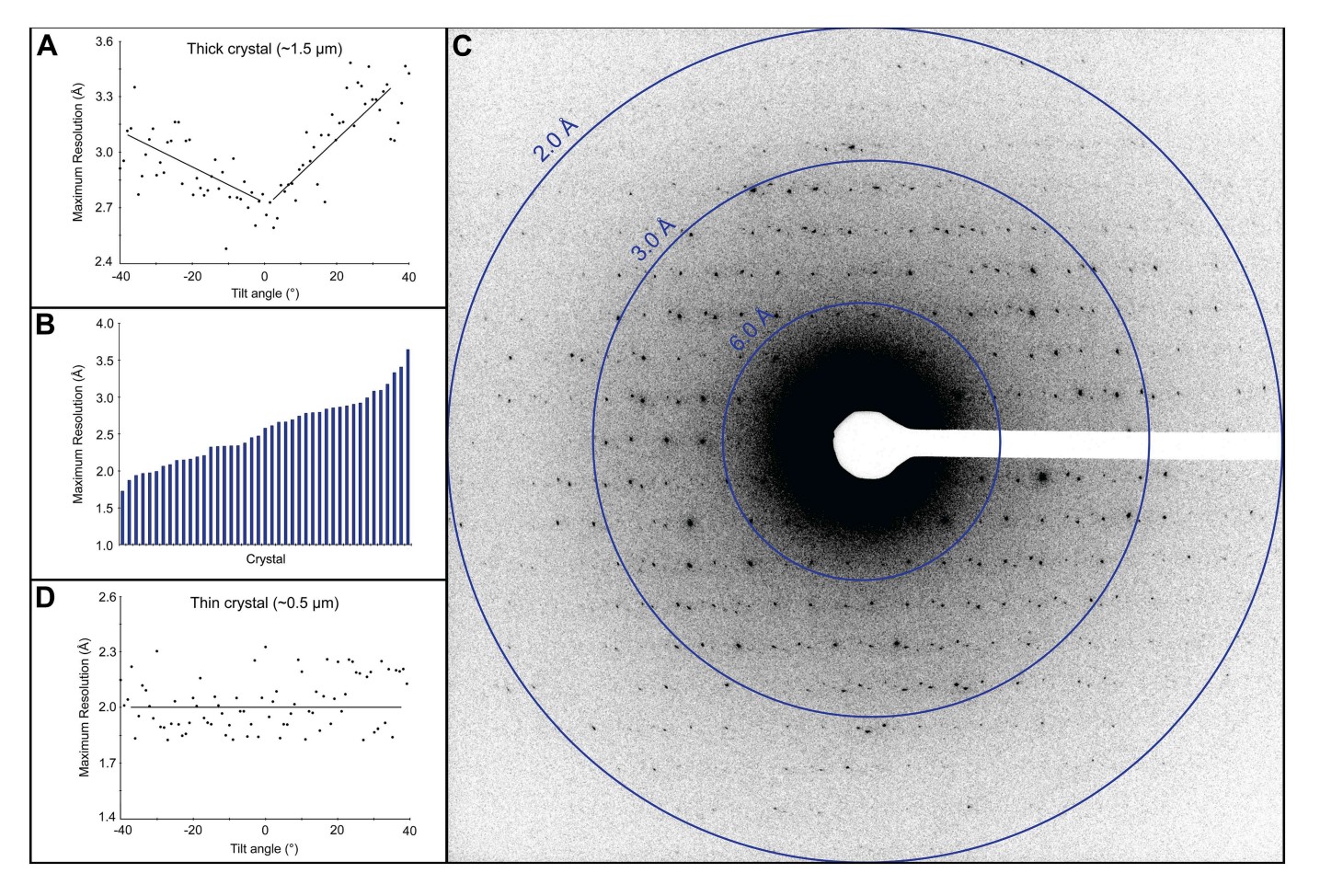

**Figure 2**. Resolution and data quality of lysozyme microcrystals. (**A**) Analysis of the effects of crystal thickness on maximum resolution of observed reflections from thick crystals. The analysis shows adverse effects of crystal thickness on the obtainable resolution as large crystals are tilted. (**B**) For assessing the quality of our cryo preparations, diffraction data were obtained from 100 lysozyme microcrystals. 43/100 were thin crystals that showed reflections in the 2–4 Å range, with the best crystal in this set yielding data to ~1.7 Å resolution. (**C**) An example of lysozyme diffraction data collected at $0.01 e^-/Å^2$/second and a 10 s exposure. The pattern shows strong and sharp spots surpassing 2 Å resolution. This diffraction pattern was processed with ImageJ and despeckled for ease of viewing. (**D**) Analysis of the effects of crystal thickness on maximum resolution of observed reflections from thin crystals. The small crystal shows a relatively constant maximum resolution that does not appear to be affected by crystal tilt.

As a dataset containing multiple exposures from a single crystal is collected, energy transferred by inelastic scattering will damage the crystalline matrix, negatively affecting both the resolution limit and intensities of observed reflections. Although the overall damage from electron scattering is much lower than that for X-rays (approximately 60 eV deposited per elastic scattering event vs 80 keV per elastic X-ray scattering event [*Henderson, 1995*]), accumulating radiation damage will eventually contribute significant error to the recorded intensities.

We performed an experiment to quantify the effects of increasing electron dosage on recorded intensities (*Figure 3*). A single protein microcrystal was subjected to sequential 10 s exposures, each delivering ~0.1 $e^-/Å^2$, until a total accumulated dose of ~12 $e^-/Å^2$ was reached. The intensities of three diffraction spots, ranging from resolutions of 2.9 to 4.6 Å, were measured on each of the 120 resulting diffraction patterns and compared. There were no observable adverse effects on resolution (*Figure 2D*) or intensity until the accumulated dose had reached ~9 $e^-/Å^2$ (*Figure 3*). We therefore optimized the data collection protocol to keep the total accumulated electron dosage below this critical value.

By using such a low dose, we could limit the radiation damage to the crystal, allowing us to collect multiple diffraction patterns from a single crystal instead of just a single pattern. Using this modified

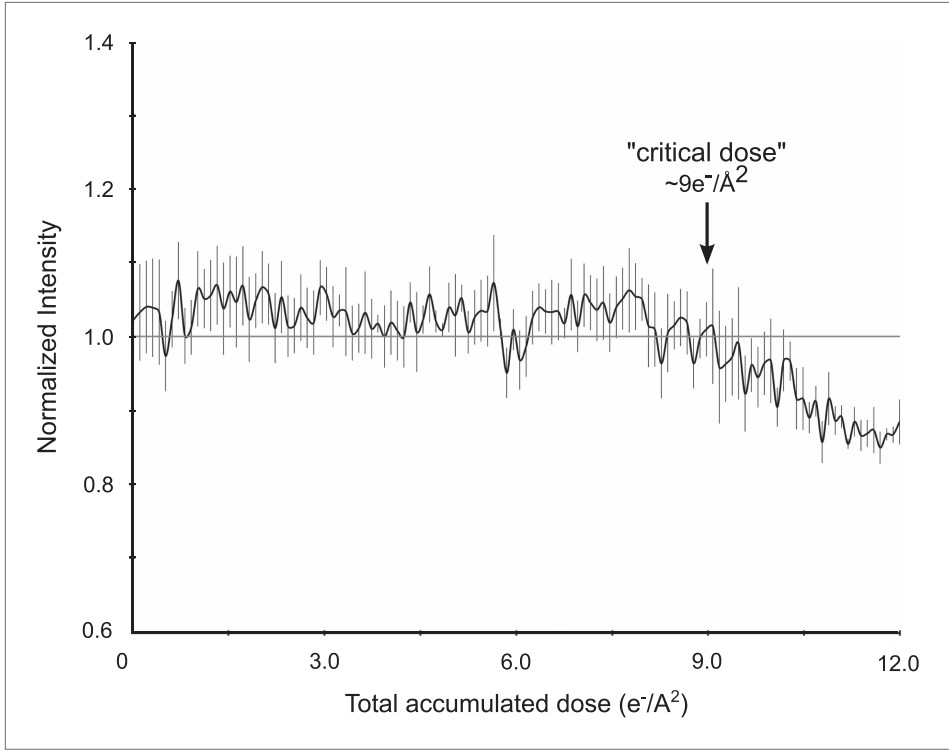

**Figure 3**. Effects of cumulative electron dose on diffraction data quality. A single lysozyme microcrystal was subjected to 120 sequential exposures without tilting, each of a dose of ~0.1 e−/Å² for a total accumulated dose of ~12 e−/Å². Normalized intensity vs total accumulated dose for three diffraction spots observed over all 120 sequential frames was plotted. A decrease in diffraction intensity becomes apparent at a dosage of ~9 e−/Å² ('critical dose'). Bars represent standard error of the mean.

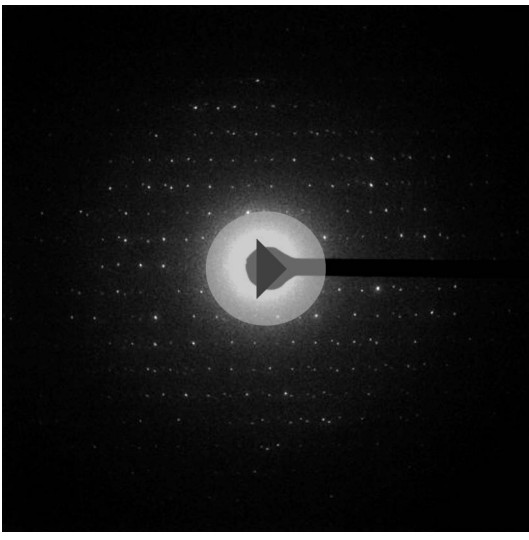

**Video 1**. An example of a complete three-dimensional electron diffraction data set from a single lysozyme microcrystal. In this example, diffraction patterns were recorded at 1° intervals from a single crystal, tilted over 47°. Cumulative dose was ~5 e−/Å² in this example.

procedure, we were able to collect up to 90 individual diffraction patterns from a single crystal (*Video 1*). Each pattern was recorded following a 1° tilt to cover ~40–90° (begin with the stage tilted at −45° and proceed to collect data to +45° in order to cover a 90° wedge). 0.1 and 0.2 degree increments were also applied to sample the reciprocal space at higher resolution. Each exposure lasted up to 10 s at a dosage of approximately 0.01 e−/Å² per second, for a cumulative dose of no more than ~9 e−/Å² per data set.

## Data processing and structure determination

The lattice parameters were determined and the lattice indexed with software based on previous studies (*Shi et al., 1998*). By collecting multiple frames from the same crystal, it was possible to determine the orientation and magnitude of the reciprocal unit cell vectors a*, b*, and c* as described in the 'Materials and methods' section. These vectors were calculated for each data set, allowing the prediction of the position of the reflections in each diffraction pattern (*Figure 4*;

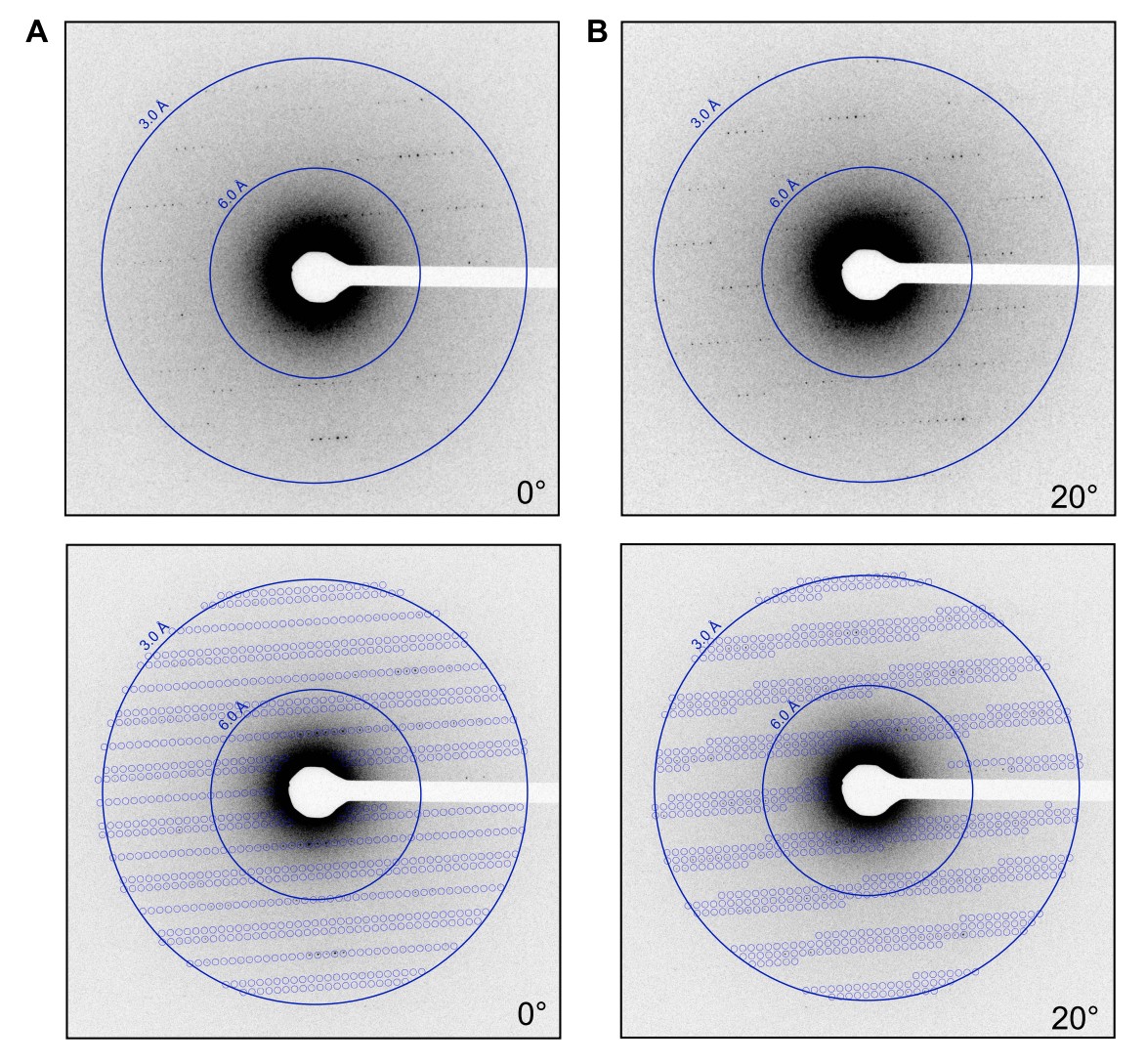

**Figure 4**. Prediction of reflections and indexing in the diffraction patterns. (**A** and **B**) Two examples of diffraction patterns obtained from a single crystal at tilt angles of 0° and 20° respectively. Locations indicated by circles were predicted to contain diffraction spots by our spot prediction algorithm. Additional examples from the same crystal are presented in *Video 2*. The resolution limit was set at 2.9 Å resolution for this study.

*Video 2*) and indexing of the entire data set. The unit cell dimensions were calculated as a = b = 77 Å, c = 37 Å, α = β = γ = 90° and P4$_3$2$_1$2 symmetry. This space group symmetry and unit cell dimensions are consistent with previous lysozyme X-ray diffraction data (*Diamond, 1974*; *Sauter et al., 2001*; *Cipriani et al., 2012*).

The electron diffraction data were collected on a microscope operating at 200 kV and equipped with a field emission gun (FEG) electron source. The FEG can generate a very coherent beam with an energy-spread function of <1 eV at 200 kV acceleration voltage. The electron beam wavelength is 0.025 Å at 200 kV compared with ~1 Å for X-rays. Under such conditions, the Ewald sphere in our experiments is nearly flat (the sphere is off the reciprocal plane by only 0.003 Å$^{-1}$ at 2 Å resolution) even in the high-resolution range. Measurements of the full width at half maximum intensity for the strongest reflections indicate that the reflections in our experiments are very tight, spreading less than a 6 pixel sphere that corresponds to ~1/1000 Å. (*Figure 5*) In our experiments, the shortest unit cell dimension for lysozyme in reciprocal space is a* = b* = 1/77 Å. Therefore, without beam oscillation or mechanical oscillation of the crystal (microscope compustage), the lattice points on a single projection that are not exactly at the Ewald sphere surface will give partial intensities.

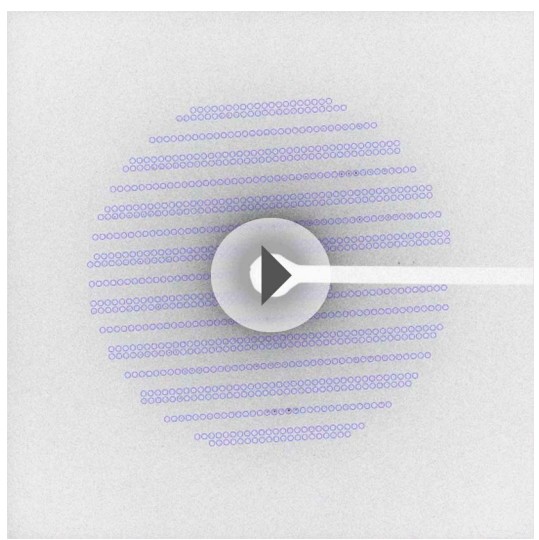

**Video 2**. An example of spot prediction in diffraction data from a single crystal. Reflections predicted on representative diffraction patterns obtained from a single crystal tilted over 39° sampled every 2° in this video. Predictions were made to 2.9 Å resolution using our spot prediction algorithm.

Because we densely sampled the reciprocal space, we recorded multiple observations for every lattice point (*Table 1*). Therefore, we could sample the observed intensity values for each reflection multiple times (multiplicity value = 34), and we made the assumption that the strongest intensity roughly approximated the complete intensity. Therefore, we kept only the maximum intensity and treated it as a unique reflection in the final structure factor file. All other recorded intensities were presumed to be partial reflections and were therefore discarded. The merging of data in P422 symmetry from three separate crystals processed in this manner resulted in a final data set with 2490 unique reflections with ~92% cumulative completeness at 2.9 Å resolution (*Table 1*, *Video 3*). The measured intensities were converted to amplitudes by assuming $I_{hkl} \approx |F_{hkl}|^2$ (*Drenth, 1994*) and an mtz file generated.

The structure of lysozyme was solved at 2.9 Å resolution by molecular replacement (MR) using the lysozyme PDB 4AXT (*Cipriani et al., 2012*) as a search model. The initial MR $2F_{obs}-F_{calc}$ map prior to refinement is presented in *Figure 6*. The map shows well-defined density around the model, indicating high quality phases from MR (*Figure 6A,B*). Likewise, a composite-omit map that was calculated by omitting 5% at a time showed good agreement with the original map obtained by MR (*Figure 6C*). When a poly-alanine (polyA) model of lysozyme was used for MR, the resulting map showed significant density beyond the alanine side chains (indicated by arrows in *Figure 6E,F*), into which the correct side chains could be built. These results indicated that our solution from MR was not dominated by model bias.

Following refinement that included the use of electron scattering factors, rigid body, simulated annealing, and B-factor refinement, a solution was found with acceptable statistics ($R_{work}/R_{free}$ = 25.5%/27.8%) and good geometry at 2.9 Å resolution (*Table 1*). The density map obtained by electron diffraction shows good agreement with the refined model (*Figure 7A*, *Video 4*). The $F_{obs}-F_{calc}$ difference map shows no interpretable features (*Figure 7B*). Additionally, the final structure has a very low RMSD (0.475 Å for Cα, 0.575 Å for all atoms) when compared to the previously published high-resolution structure of lysozyme (*Cipriani et al., 2012*).

## Model validation and bias tests

To further validate the method and test for model bias, we performed a number of tests on the data to check whether a good solution could be obtained from random noise as has been demonstrated for electron micrographs (*Shatsky et al., 2009*). We created multiple randomized datasets to test the robustness of the phasing and model building procedure. The test datasets were generated as follows:

1. All measured intensities were replaced with random numbers ranging between the minimum and maximum of the actual observed experimental values.
2. The experimental intensity values were kept but the Miller indices were randomized.
3. All experimental intensities were replaced with an actual intensity value that was measured by X-ray crystallography of an unrelated structure (Calmodulin PDB ID:3SUI [*Lau et al., 2012*]).
4. Each experimental intensity was increased or decreased randomly by up to 35%.

In addition, the correct experimental dataset was also used and labeled as dataset '5'. These five datasets were treated as 'blind test cases', in which the user did not know the identities of the various test datasets. Each test dataset was used for molecular replacement with the lysozyme model (*Cipriani et al., 2012*), followed by a single round of refinement in PHENIX (*Adams et al.,*

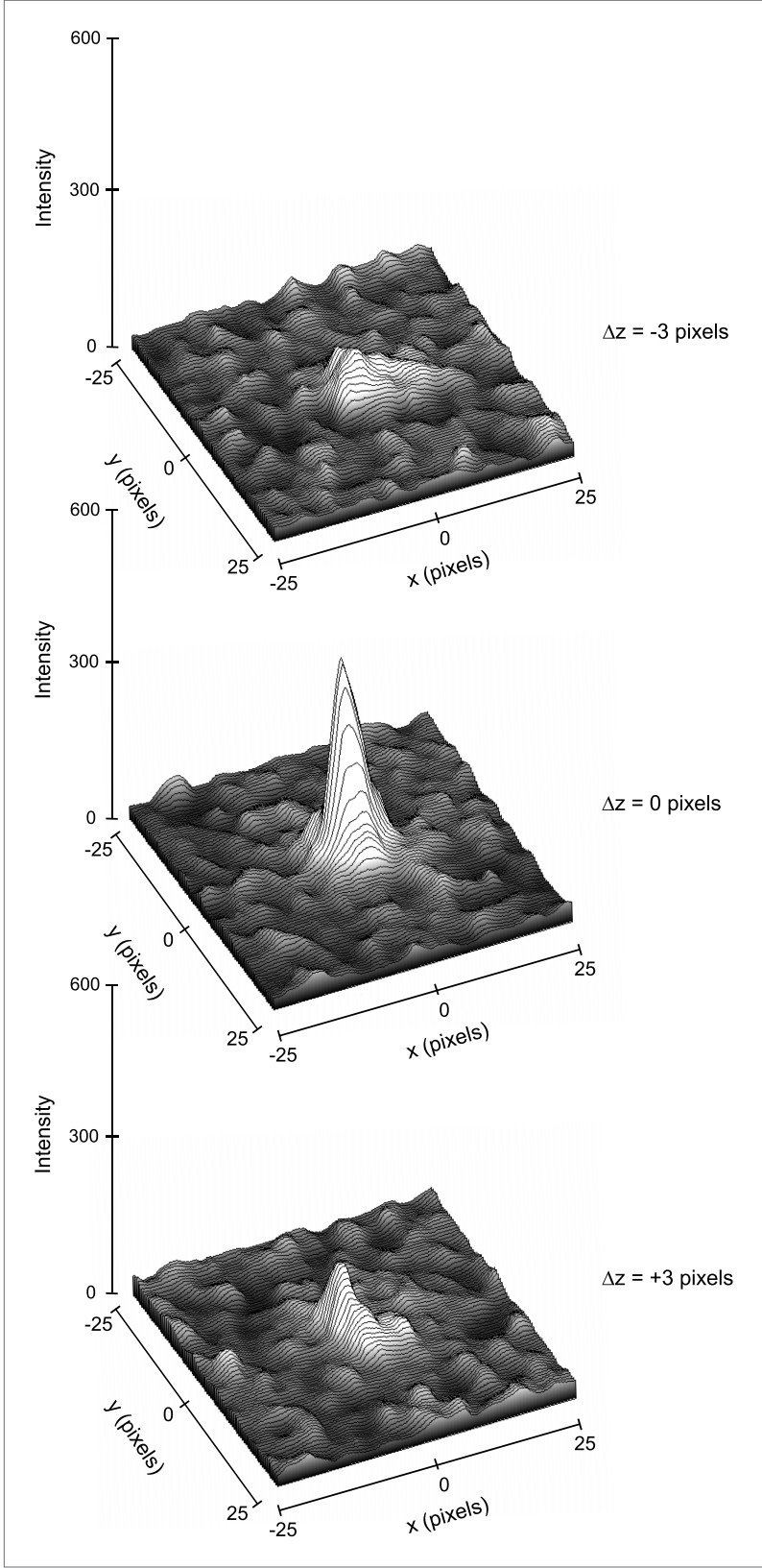

**Figure 5**. Three-dimensional profiles of the intensity of a single reflection over three consecutive diffraction patterns at −0.1°, 0°, and 0.1° degree tilts. The plots show the approximate dimensions of the full reflection with a width (full width at half maximum height) of 3–5 pixels in the x, y, and z direction.

**Table 1.** MicroED crystallographic data

| Data collection | |
|---|---|
| Excitation voltage | 200 kV |
| Electron source | Field emission gun |
| Wavelength (Å) | 0.025 |
| Total electron dose per crystal | ~9 e$^-$/Å$^2$ |
| Number of patterns per crystal | 40–90 |
| No. crystals used | 3 |
| Total reflections to 2.9 Å | 84,889 |
| Data refinement | |
| Space group | P4$_3$2$_1$2 |
| Unit cell dimensions | |
| a = b | 77 Å |
| c | 37 Å |
| α = β = γ | 90° |
| Resolution | 2.9–20.0 Å |
| Total unique reflections | 2490 |
| Reflections in working set | 2240 |
| Reflections in test set | 250 |
| Multiplicity* | 34 |
| Completeness (2.9–3.1) | 92% (57%) |
| R$_{work}$/R$_{free}$ (%) | 25.5/27.8 |
| RMSD bonds | 0.051 Å |
| RMSD angles | 1.587° |
| Ramachandran (%)† (allowed, generous, disallowed) | 99.1; 0.9; 0 |

*Multiplicity is defined as total measured reflections divided by number of unique reflections.
†Statistics given by PROCHECK (*Laskowski et al., 1993*).

*2010*). Only dataset 5, which contained the correct observed experimental intensities, yielded a solution that could be further refined to acceptable R$_{work}$/R$_{free}$ and geometry. Datasets 1–4, which contained the random errors described above, either did not yield MR solutions or would not allow refinement to produce an acceptable structure (*Table 2*).

We also tested the robustness of the MR procedure by using a number of unrelated structures, chosen from the PDB for their similar unit cell dimensions and protein molecular weights, as search models against our experimental data. The unrelated structures were: T4 lysozyme, calmodulin, dodecin, and αA crystallin (*Table 3*). None of these structures gave an acceptable MR solution.

Together, these experiments indicate that the extracted intensities are accurate enough to yield a reliable structure and that model bias originating from MR did not skew our results.

## Completeness and accuracy of the measured intensities in electron diffraction

Data sets collected in electron crystallography of 2D crystals suffer from a missing cone due to the limitation of the maximum achievable tilt angle in the TEM. Previous reports estimate that with tilt angles up to 60°, the missing cone is roughly 13% (*Glaeser et al., 1989*), and the resolution in plane is typically higher than the resolution perpendicular to the tilt axis (z*). In our experiments, because the data from 3 crystals were eventually used, and the orientation of each crystal on the grid varied, we could cover the full reciprocal space (*Video 3*).

Dynamic scattering likely introduces inaccuracies in the electron diffraction data. In electron diffraction, dynamic scattering (multi scattering events) could redistribute primary reflection intensities, reducing the accuracy of the intensity measurements by randomly contributing to the observed intensities (*Grigorieff et al., 1996*). The lysozyme crystals have P4$_3$2$_1$2, symmetry and systematic absences are expected at (2n+1,0,0). However, very weak reflections were observed at the positions where absences were expected (*Figure 8*). It is likely that these reflections originate from dynamic scattering events. We plotted the intensities along the a* and b* axes and compared the intensity values. The intensities of Miller indices (2n+1,0,0) and (0,2n+1,0) were measured and compared to the intensities of the four immediately adjacent reflections (2n+2,1,0), (2n+2,−1,0), (2n−2,1,0), and (2n−2,−1,0). On average, the intensity in the systematic absences was found to be 4.9% of the total intensity of the adjacent spots. (Standard deviation 2.7%, Max 12.4%, n = 17). Moreover, comparison of our experimental intensities with intensities obtained by X-ray diffraction of lysozyme of the same crystal form indicates that our data follow a similar trend and are not dominated by intensity randomness. A Pearson correlation coefficient between the two data sets was 0.63 from 6.0 to 13.5 Å (0.56 from 2.0–13.5 Å), indicating conservation of reflection hierarchy—strong intensities remain strong and weak intensities remain weak. Together, our analyses suggest that multiple scattering contributes at maximum roughly 10% to the intensity value and that at least for structure determination at 2.9 Å resolution such an error in intensity appears to be tolerable. It is possible that dynamic scattering will become a significant source of error at higher resolutions and some correction algorithm will then have to be developed.

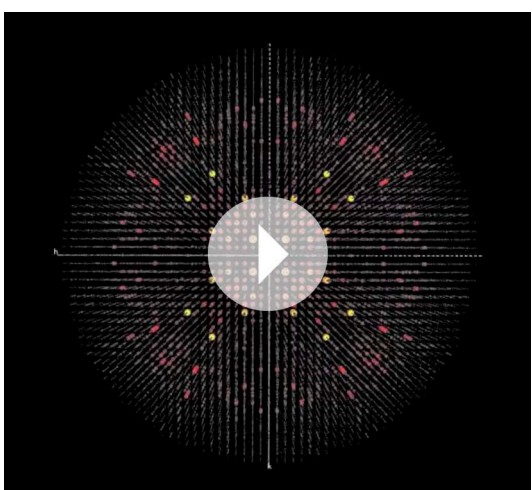

**Video 3**. Three-dimensional representation of merged intensity values. 2490 total unique reflections are present for an overall completeness of 92% at 2.9 Å resolution. Video begins with a* axis horizontal, b* axis vertical, and the c* axis normal to the image plane.

## Discussion

We present a method, 'MicroED', for structure determination by electron crystallography. It should be widely applicable to both soluble and membrane proteins as long as small, well-ordered crystals can be obtained. We have shown that diffraction data at atomic resolution can be collected and a structure determined from crystals that are up to 6 orders of magnitude smaller in volume than those typically used for X-ray crystallography.

For difficult targets such as membrane proteins and multi-protein complexes, screening often produces microcrystals that require a great deal of optimization before reaching the size required for X-ray crystallography. Sometimes such size optimization becomes an impassable barrier. Electron diffraction of microcrystals as described here offers an alternative, allowing this roadblock to be bypassed and data to be collected directly from the initial crystallization hits.

While our proof of principle is an important first step, further optimization of the method is required. Better programs need to be developed for accurately determining lattice parameters, indexing all reflections, extracting the intensities and correcting for incomplete intensities, dynamic scattering, and Ewald sphere curvature. Specifically, developing procedures for postrefinement (unit cell refinement, estimating the mosaic spread, rocking curve, etc) should allow for the proper correction and scaling of partially recorded reflections, leading to improved estimation of full intensities. Relatively minor modifications to existing programs such as MOSFLM (*Leslie and Powell, 2007*) should allow the handling of electron diffraction data from 3D crystals and take advantage of the large body of work already dedicated to processing X-ray diffraction data.

The accuracy of the microscope compustage can be improved and procedures for crystal or beam oscillation implemented. Our method of using the maximum intensity measurement as an approximation of the full intensity of any given spot is admittedly crude, as it depends on the intersection of the Ewald sphere through the center of each spot at some point in the tilt series. As the resolution increases, this event becomes increasingly unlikely. Crystal oscillation or related methods such as precession of the electron beam (*Gjønnes et al., 1998*) would allow more accurate determination of spot intensities, especially at very high resolutions.

Further development of various methods for phasing the diffraction data are also required and could possibly include heavy metal phasing. Such phasing methods are standard in X-ray crystallography and rely on differences in intensity values between a native data set and heavy metal derivative data sets. It is possible that in electron crystallography dynamic scattering could hinder phasing by such methods, and new algorithms will need to be developed to make this possible. Phase extension from projection maps or from low-resolution density maps can also be used for direct phasing (*Gipson et al., 2011*; *Wisedchaisri and Gonen, 2011*). It is also possible that single particle cryo-EM could be used for direct phasing as previously demonstrated where a low-resolution single particle map was used to phase X-ray diffraction data (*Speir et al., 1995*; *Dodson, 2001*; *Xiong, 2008*). Moreover, a double tilt cryo holder as well as newly developed goniometer-based grid holders could be used to cover more of the Fourier space. Finally this method could benefit from automation in data collection.

This first study serves as a proof of principle that three-dimensional electron diffraction can yield an accurate protein structure from microcrystals. As additional protocols and programs are developed, MicroED promises to advance the field of structural biology and open the door to many exciting new studies.

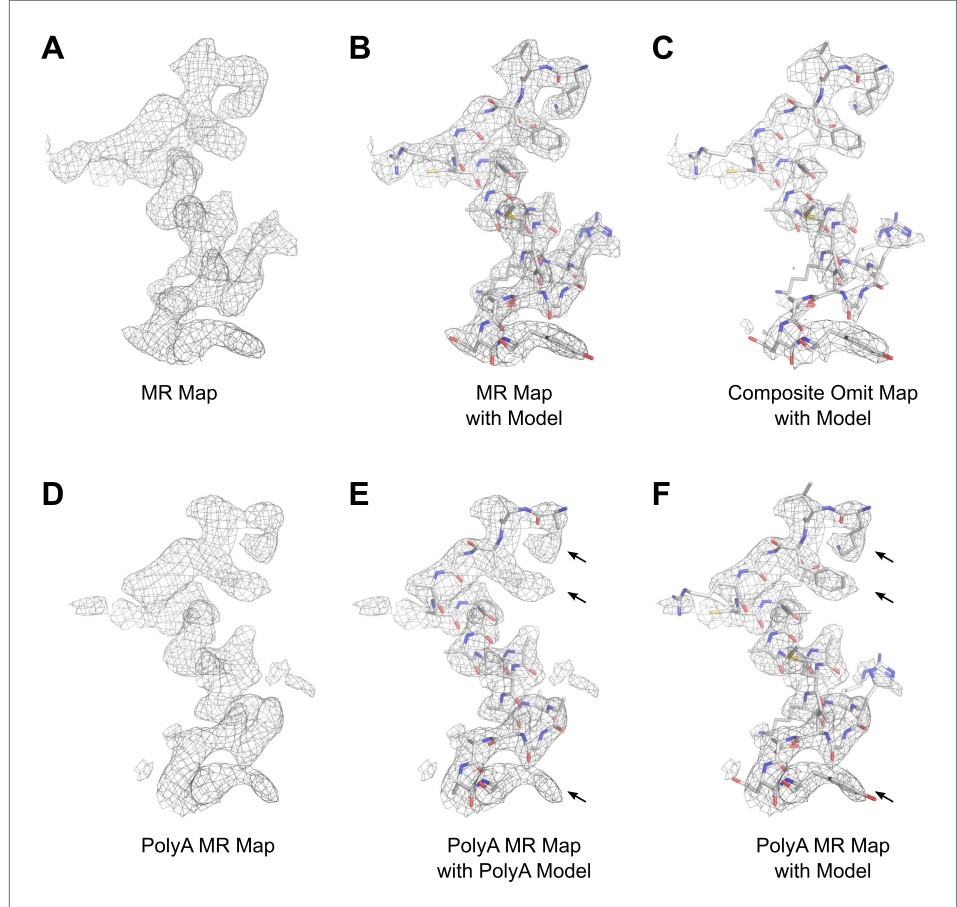

**Figure 6**. Results of phasing by molecular replacement prior to crystallographic refinement. Molecular replacement was performed with both the full model of lysozyme (PDB 4AXT, top panels) as well as a poly-alanine model (bottom panels) and the resulting $2F_{obs}-F_{calc}$ maps around residues 1–20 are shown. (**A** and **B**) The phases following molecular replacement with the full model were of good quality demonstrated by how well the density surrounding the model fits, even before any refinement is performed. (**C**) A composite-omit map calculated by omitting 5% at a time showed good agreement with the unrefined structure indicating the phases were not dominated by model bias. (**D**–**F**) As an additional test of model bias, phasing was done with a poly-alanine homology search model of lysozyme. The resulting $2F_{obs}-F_{calc}$ map is of good quality (**D**) and shows density extending beyond the poly-alanine model (**E** and **F**, arrows). (**F**) The same density map as **E** but with the structure of lysozyme fit. Arrows in **D** and **E** show examples of clear side chain density from the poly-alanine map. All maps are contoured at 1.0σ.

## Materials and methods

### Lysozyme crystallization and sample preparation
Lysozyme was purchased from Fisher Scientific and a 200 mg/ml solution was prepared in 50 mM sodium acetate pH 4.5. Lysozyme solution was mixed 1 to 1 with precipitant solution (3.5M sodium chloride; 15% PEG 5,000; 50 mM sodium acetate pH 4.5) and crystals were grown by the hanging drop method. Following the crystal formation, the sample was diluted three to five times in 5% PEG 200. A 5 µl drop of the crystal solution was applied to a quantifoil 2/2 holey-carbon copper EM grid. The grid was then blotted and vitrified by plunging into liquid ethane using a Vitrobot Mark IV (FEI). The frozen-hydrated grid was loaded onto a Gatan 626 cryo-holder and transferred to a cryo-TEM.

### Electron diffraction
All electron microscopy was performed on a FEI Tecnai F20 TEM equipped with a field emission electron source (FEG) and operating at 200 kV. Electron diffraction pattern tilt series data were recorded

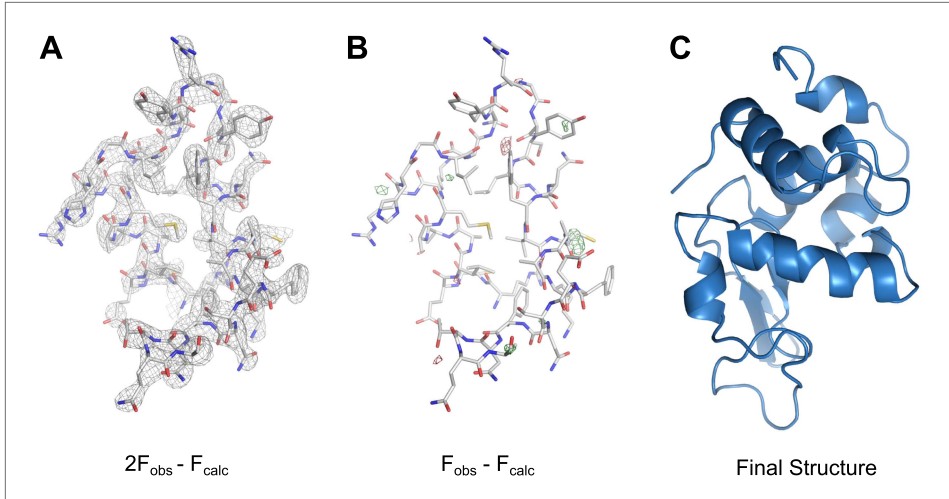

**Figure 7**. MicroED structure of lysozyme at 2.9 Å resolution. (**A**) The $2F_{obs}-F_{calc}$ (contoured at 1.5σ) map covers protein residues 5–45 of lysozyme. (**B**) $F_{obs}-F_{calc}$ difference map contoured at +3.0σ (green) and −3.0σ (red) for the same protein region. The map (**A**) shows well-defined density around the vast majority of side chains and the difference map (**B**) shows no large discrepancies between the observed data ($F_{obs}$) and the model ($F_{calc}$). The final structure of lysozyme is shown in panel **C** and the complete three-dimensional map is presented in *Video 3*.

with a bottom mount TVIPS F416 4 k × 4 k CMOS camera with pixel size 15.6 μm using built in series exposure mode. The electron dose was kept below 0.01 e⁻/Å² per second, and each frame of a data set was taken with an exposure time of up to 10 s per frame. The electron dosage was calibrated with the use of a Faraday cage as well as by calibrating the counts on the CMOS detector in bright field mode. Each data set consisted of up to 90 still frames taken at 0.1–1° intervals with a maximum total dose of ~9e⁻/Å² per crystal. The camera length was optimized for the desired resolution as described previously (*Gonen, 2013*).

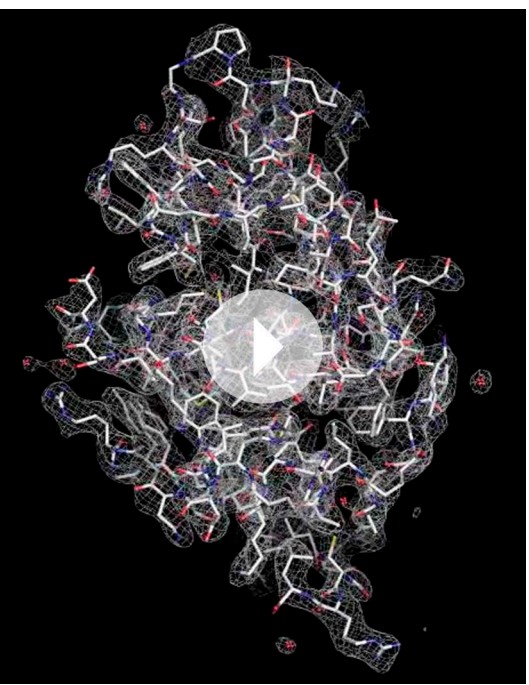

**Video 4**. $2F_{obs}-F_{calc}$ density around the complete lysozyme model at 2.9 Å resolution (contoured at 1.5σ).

## Data processing

Although our original intent was to perform all data analysis with existing X-ray crystallography software various incompatibilities and logistical roadblocks necessitated the development of some additional tools. Diffraction patterns were indexed and background subtracted intensities extracted and merged with in-house developed software implemented in python using methods adapted from those developed by *Shi et al. (1998)*.

Briefly, measurements were made on images identified as major planes of the crystal with ImageJ and used to determine the approximate magnitudes of the unit cell vectors a*, b*, and c* and the angles between them (α, β, and γ). Subsequently, 100 to 350 spots were chosen across several images from each set of diffraction patterns. Vectors in reciprocal space were calculated for all of the selected spots. Difference vectors between spot vectors were calculated allowing vectors approximating the estimated unit cell lengths to be identified. The angles between potential unit cell vectors were calculated

**Table 2.** Results of model validation and bias tests

| Data set | Molecular replacement result | TFZ | Final $R_{free}$ (%)¶ |
|---|---|---|---|
| 1* | No solution | N/A | N/A |
| 2† | Solution** | 19.1 | 54.9 |
| 3‡ | No solution | N/A | N/A |
| 4§ | Solution | 12.6 | 35.2 |
| 5# | Solution | 14.7 | 27.8 |

*Random intensities.
†Shuffled Miller indices.
‡Calmodulin replaced intensities.
§Intensities ± 35%.
#Original data.
¶Final $R_{free}$ after a minimum of two cycles of refinement.
**Solution was found; however, the space group was incorrect ($P4_12_1$).

and 'orthogonal triplets' identified. Orthogonal triplets are defined as sets of vectors that contain a predicted a*, b*, and c*, which are all 90° from each other ($\alpha = \beta = \gamma = 90°$ for this crystal). All sets of the orthogonal triplets were averaged to yield estimated a*, b*, and c* vectors. The estimated a*, b*, and c* vectors were then refined by identifying parallel difference vectors derived from the original selected spots with lengths that were multiples of the unit cell lengths.

The calculated unit cell vectors were then used to predict the spots in each diffraction pattern. Two reference spots were chosen for each image and their Miller indices calculated using the previously

**Table 3.** Models for molecular replacement validation

| Protein | PDB ID | Molecular weight (kDa) | Symmetry | Unit cell dimensions | MR solution |
|---|---|---|---|---|---|
| Hen Egg White Lysozyme* | 4AXT | 14.3 | $P4_32_12$ | a = b = 78.24 Å | Yes |
| | | | | c = 37.47 Å | |
| | | | | $\alpha = \beta = \gamma = 90°$ | |
| T4 Lysozyme† | 2LZM | 18.7 | $P3_212$ | a = b = 61.20 Å | No |
| | | | | c = 96.80 Å | |
| | | | | $\alpha = \beta = 90°$ | |
| | | | | $\gamma = 120°$ | |
| Calmodulin‡ | 3CLN | 16.7 | P1 | a = 29.71 Å, | No |
| | | | | b = 53.79 Å, | |
| | | | | c = 24.99 Å | |
| | | | | $\alpha = 94.13°$, | |
| | | | | $\beta = 97.57°$, | |
| | | | | $\gamma = 89.46°$ | |
| Dodecin§ | 4B2J | 8.5 | $F4_132$ | a = b = c = 142.90 Å | No |
| | | | | $\alpha = \beta = \gamma = 90°$ | |
| αA Crystallin# | 3L1E | 11.9 | $P4_12_12$ | a = b = 56.22 Å, | No |
| | | | | c = 68.66 Å | |
| | | | | $\alpha = \beta = \gamma = 90°$ | |

*Cipriani et al. (2012).
†Weaver and Matthews (1987).
‡Babu et al. (1988).
§Staudt et al. (2013).
#Laganowsky et al. (2010).

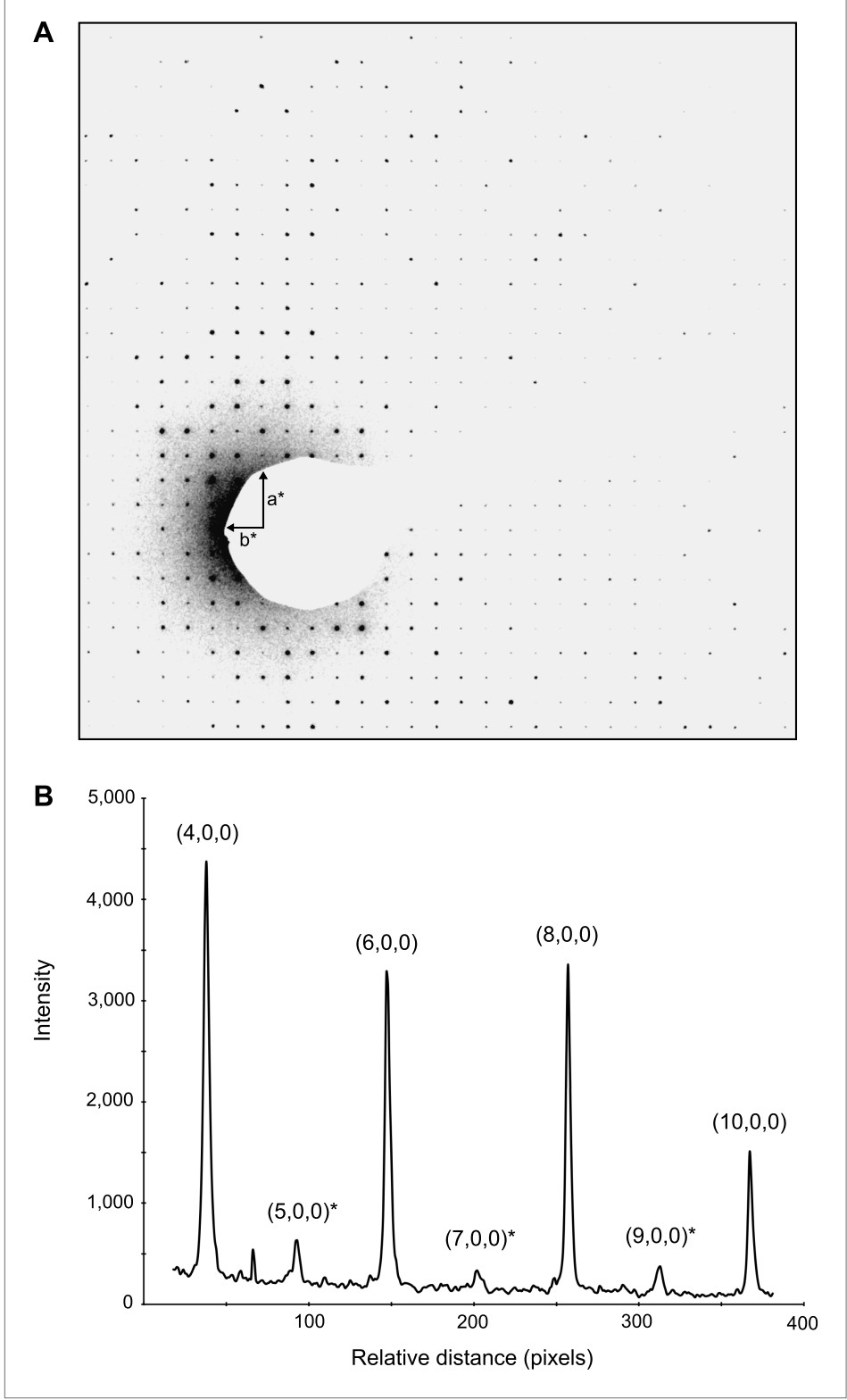

**Figure 8**. Dynamic scattering in lysozyme 3D crystals. Intensity measurement along the a* axis of a raw diffraction pattern illustrating the relatively small contributions due to dynamic scattering. (**A**) Diffraction pattern from the major plane of a lysozyme crystal with visible intensity in the (2n+1,0,0) and (0,2n+1,0) Miller indices. (**B**) (2n+1, 0, 0) reflections (starred) are expected to be systematically absent and observed intensities at these indices are assumed to be the result of dynamic scattering. Image contrast was enhanced for clarity using ImageJ.

determined unit cell vectors. For every diffraction pattern, the vector normal to the detector plane was calculated as:

$$\mathbf{r}_1 \times \mathbf{r}_2 = \mathbf{n}$$

where $\mathbf{r}_1$ and $\mathbf{r}_2$ are the vectors defined by the Miller indices from reference spots one and two, respectively, and $\mathbf{n}$ is the resulting vector normal to the detector plane. Any reflection that appears on a given diffraction pattern will satisfy:

$$\mathbf{n} \cdot \mathbf{v} = 0$$

where $\mathbf{v}$ is any set of Miller indices. For any h, k, l that satisfied the above equation, within a defined threshold, that particular reflection was predicted to appear on the diffraction image, and its x, y detector coordinates on the diffraction pattern image were calculated.

Intensities for each predicted reflection were integrated by first drawing both a square and a circular mask centered on the reflection, with the diameter of the circle identical to the length of the square. The mean pixel intensity outside the circle but within the square was calculated yielding the mean background intensity. The mean background was then subtracted from each pixel within the circle, and the resulting pixel intensities were summed. All related intensities from three data sets were grouped based on P422 symmetry. The maximum value for each group of equivalent reflections was assumed to best approximate the full intensity and was used for that reflection in the final data set. Because each intensity measurement ultimately originated from a single observation, SigI and SigF values were estimated as the square root of the intensity and square root of the structure factor, respectively. The final mtz file contains columns h, k, l, F, SIGF, I, SIGI. The final data set contained 2490 unique reflections from 2.9–20 Å with cumulative completeness of 92% (*Table 1*).

## Structure refinement

Phaser (*McCoy et al., 2007*) was used to obtain phases with lysozyme structure 4AXT (*Cipriani et al., 2012*) as a MR search model (LLG = 372 and TFZ = 14.7). The structure was then refined using CNS (*Brünger et al., 1998*) and PHENIX (*Adams et al., 2010*) by rounds of rigid body, simulated annealing, and B-factor refinement. The $R_{free}$ data set represented 10% of the total data set. The data were subjected to twinning analysis; however, twinning with this symmetry group is forbidden and therefore we ruled out twinning in our crystals. Electron scattering factors (*Gonen et al., 2005*) were used during refinement.

## Data deposition and software availability

The structure factors and coordinates of the final model were deposited in the Protein Data Bank with accession code 3J4G. The in house developed program that was used for processing the MicroED data is available for download at http://www.github.com/gonenlab/microED.git.

## Acknowledgements

The authors wish to thank members of the Gonen lab and Goragot Wisedchaisri for helpful discussions. We also would like to thank Nikolaus Grigorieff (HHMI, Janelia Farm Research Campus) for helpful discussions and suggestions and for critically reading this manuscript. Shane Gonen (JFRC, UW) assisted with the preparation of *Video 3*. This work is dedicated to the memory of Prof K H Kuo.

## Additional information

### Funding

| Funder | Author |
| --- | --- |
| Howard Hughes Medical Institute | Dan Shi, Brent L Nannenga, Matthew G Iadanza, Tamir Gonen |

The funder had no role in study design, data collection and interpretation, or the decision to submit the work for publication.

### Author contributions

DS, BLN, MGI, TG, Conception and design, Acquisition of data, Analysis and interpretation of data, Drafting or revising the article

# Additional files

## Major dataset

The following dataset was generated:

| Author(s) | Year | Dataset title | Dataset ID and/or URL | Database, license, and accessibility information |
|---|---|---|---|---|
| Shi D, Nannenga B.L, Iadanza M.G, Gonen T | 2013 | Structure of lysozyme solved by MicroED to 2.9Å | 3J4G; http://www.rcsb.org/pdb/explore/explore.do?structureId=3j4g | Publicly available at RCSB Protein Data Bank (http://www.rcsb.org). |

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
