## [Decision Letter]

Thank you for sending your work entitled “Three-dimensional electron crystallography of protein microcrystals” for consideration at *eLife*. Your article has been favorably evaluated by a Senior editor and 3 reviewers.

The following individuals responsible for the peer review of your submission have agreed to reveal their identity: Stephen Harrison (guest Reviewing editor), Richard Henderson, and Axel Brunger (peer reviewers).

The guest Reviewing editor and the other reviewers discussed their comments before we reached this decision, and the guest Reviewing editor has assembled the following comments to help you prepare a revised submission.

This is an interesting paper reporting a successful, 2.9Å resolution, electron crystallographic study of tetragonal lysozyme crystals. It must be viewed in the context of the publication 50 years ago of the 6Å X-ray structure of the same tetragonal lysozyme crystals (Blake et al., Nature, 1962), the 2Å structure 3 years later (Blake et al., Nature, 1965) and the 0.94Å structure in 2001, which is cited in the present paper. Lysozyme has been a prototype for testing out new methods and approaches ever since Roberto Poljak crystallized it in the 1950s. It has also been studied recently by electron diffraction and imaging (Nederhof et al., Acta Cryst D69, 1223-1230 (2013); Nederhof et al., Acta Cryst D69, 852-859 (2013)). It would be helpful if the authors of the current paper cited the above publications. There is also Moukhametzianov et al., Acta Cryst D, 64, 158-166 (2008) who collected a full 3D 1.5Å resolution dataset from very small microcrystals of xylanase using an X-ray microfocus beam.

Although the authors have done well to navigate all the technical hurdles that have so far prevented anyone else from collecting a full set of 3D electron diffraction data, it is still not clear whether there is any practical advantage in using electrons over conventional X-rays for 3D crystals. As Moukhametzianov et al., showed, there is also less radiation damage in X-ray crystallography when the crystals get very small because the secondary photoelectrons that are emitted when an X-ray is adsorbed can escape without depositing all their energy inside the crystal. The central background of inelastically scattered electrons in the electron diffraction patterns the present authors show is quite high. This high background could be greatly improved by using a zero-loss energy filter to remove these electrons. The paper should discuss the issue of damage, include some damage statistics from the present work (see item 2 in the revision requests, below), and mention the longer-term value of using a zero-loss energy filter.

The authors have also used a very crude way to measure intensities, simply taking the highest intensity of each peak in the rotation series. This would be greatly improved if the now fully established method of profile fitting that is used by the X-ray crystallographers and the methods for correcting partial reflection suggested below were implemented. (See changes in Abstract requested below.) If the work could be repeated with zero-loss energy filtering, a better detector, plus proper estimation of intensities, we might then be in a better position to see how successful the technique is, and what potential it has. Nevertheless, this is a useful step along the way.

We recommend two changes in the Abstract.

1) Change the first sentence to: “We demonstrate that it is feasible to determine high-resolution proteins structures by electron crystallography of three-dimensional crystal in an electron cryo-microscope (CryoEM).” The current first sentence is a trace misleading, because the study is really a demonstration of feasibility rather than a de novo structure determination.

2) Change the last sentence to: “This proof of principle paves the way for the implementation of a new technique that may have wide applicability in structural biology”.

The following substantive points should be addressed in a revision.

1) Specify the significance of the MR solution (TFZ and LLG). A 2Fo-Fc map should be shown after MR in order to assess the quality of the MR phases. Moreover, a “omit” refinement should be performed with a small fragment of the protein omitted (< ∼10%) and a difference map of that omitted region calculated in order to assess the potential model bias of the maps.

2) Plot a quantity related to radiation damage (for example, the average diffraction intensity) as a function of dose for each subsequent shot for the three crystals used for the final data set.

3) Figure 2: Is the diffraction limit of the crystals related to size? In other words, is the useful limit related to the number of electrons per Bragg peak?

4) There is an XFEL lysozyme structure available to higher resolution (to dmin = 1.9) (Boutet et al., Science 2012) with comparable crystal size (less than 1 micrometer by 1 micrometer by 3 micrometers). This perhaps suggests that low-dose electron diffraction data collection is ultimately limited by radiation damage or the number of electrons per Bragg peak, in contrast to XFELs where radiation damage is circumvented, even at high-photon flux. The authors should comment on this.

5) Did the authors try a standard indexing programs used in X-ray crystallography? It is curious why a new indexing program was developed.

6) Taking the maximum intensity among the set of equivalent reflection in order to approximate the fully recorded refection is a very crude approximation that depends on the chance of recording approximately fully recorded reflections. This chance in turn is related to the energy bandwidth of the electron beam, the crystal mosacity, crystal and detector parameters, and crystal size (for very small crystals). A discussion of these factors and parameters for this particular experiment would be useful.

7) The “blind” test cases either did not lead to a MR solution or they would not allow refinement. Please specify details: did some of the blind tests produce an MR solution, but refinement was stuck at high Rfree?

8) The observation of non-zero intensities for systematic extinctions is interesting. The explanation by dynamic scattering is possible, although could there be other possibilities, e.g., due to small crystal size?

9) In the Introduction the authors commented on XFELs: the large sample requirement is a limitation of current delivery methods and data analysis (i.e., without post-refinement), not due to XFELs per se. However, the comment on the limited availability of beam time is well taken. In this respect, the low-dose method presented by the authors could be an important staging method for XFEL experiments, and the combination of both experiments could be very powerful.

---

## [Author Response]

*The following substantive points should be addressed in a revision*.

*1) Specify the significance of the MR solution (TFZ and LLG). A 2Fo-Fc map should be shown after MR in order to assess the quality of the MR phases. Moreover, a “omit” refinement should be performed with a small fragment of the protein omitted (< ∼10%) and a difference map of that omitted region calculated in order to assess the potential model bias of the maps*.

The significance of the MR solution is indicated in the methods section and a new figure was added showing the 2Fo-Fc map after MR and a composite omit map (5% omit) (Figure 6). These maps are also discussed in the revised text.

*2) Plot a quantity related to radiation damage (for example, the average diffraction intensity) as a function of dose for each subsequent shot for the three crystals used for the final data set*.

For each crystal that was used, we planned to plot average intensity versus dose at the resolution shell of 3-4Å. However, once we began the analysis we found that the large number of partial intensities made it difficult to accurately track any individual spot over a complete data set of up to 90 frames. We therefore performed an additional experiment to quantify the effect of dose on spot intensities. Diffraction data was collected from a single lysozyme microcrystal using a dose of 0.1e-/Å2 per frame. 120 frames were collected without tilting the crystal for a total accumulated dose of ∼12e-/Å2. In this way we could follow individual reflections throughout all frames and plotted the intensity values versus accumulated dose. This new data is presented as Figure 3 and discussed in the revised text. We found that the diffraction intensity is not adversely affected until the accumulated dose reaches a “critical” value of ∼9e-/Å2.

*3)*
Figure 2*: Is the diffraction limit of the crystals related to size? In other words, is the useful limit related to the number of electrons per Bragg peak*?

We do not believe this is the case as even crystals significantly smaller than those used here contain more than enough unit cells to get strong Bragg peaks. We updated the text to reflect this point more clearly.

A major limitation that we observed is likely because of inelastic scattering in thick crystals. We believe that we are not limited by the number of electrons per Bragg peak but instead inelastic scattering from crystals that are too thick is severely limiting the data. In this revised version of our manuscript we added data demonstrating that the achievable resolution is affected by crystal thickness. These are presented in Figure 2 and described in the revised text.

*4) There is an XFEL lysozyme structure available to higher resolution (to dmin = 1.9) (Boutet et al., Science 2012) with comparable crystal size (less than 1 micrometer by 1 micrometer by 3 micrometers). This perhaps suggests that low-dose electron diffraction data collection is ultimately limited by radiation damage or the number of electrons per Bragg peak, in contrast to XFELs where radiation damage is circumvented, even at high-photon flux. The authors should comment on this*.

We note that data in our study was collected to 1.7Å resolution but that the structure was determined to 2.9Å resolution. This resolution cut off is self imposed based on our current software limitations. As such we note that the resolution collected here is comparable to the study by Boutet et al. Science 2013. We updated the manuscript at various places (including the Abstract) to make this clearer. We apologize for the confusion and for not making this point clearer in the original version of our manuscript.

*5) Did the authors try a standard indexing programs used in X-ray crystallography? It is curious why a new indexing program was developed*.

We did initially attempt to use MOSFLM to process our data but were met with several roadblocks that, although not insurmountable, made it more practical to develop some basic programs of our own for this initial proof of principal. We still believe that one of the future improvements of this work will be its integration into current data processing software used in X-ray crystallography. Text was added to the Discussion and Methods sections highlighting this point.

*6) Taking the maximum intensity among the set of equivalent reflection in order to approximate the fully recorded refection is a very crude approximation that depends on the chance of recording approximately fully recorded reflections. This chance in turn is related to the energy bandwidth of the electron beam, the crystal mosacity, crystal and detector parameters, and crystal size (for very small crystals). A discussion of these factors and parameters for this particular experiment would be useful*.

We added additional text to the Discussion to highlight these points and suggested how some future improvements to data collection may improve the accuracy of the measured intensities.

*7) The “blind” test cases either did not lead to a MR solution or they would not allow refinement. Please specify details: did some of the blind tests produce an MR solution, but refinement was stuck at high Rfree*?

Table 2 was added to provide this information.

*8) The observation of non-zero intensities for systematic extinctions is interesting. The explanation by dynamic scattering is possible, although could there be other possibilities, e.g., due to small crystal size*?

We do not believe that the crystal size played a role here. Extensive studies from material science describe systematic absences appearing in diffraction patters dominated by dynamic scattering. When precession data was collected these spots were averaged out (For example see references Gemmi and Nicolopoulous, (2007) Ultramicroscopy 107:483-494; [14] Acta Cryst A 54: 306-319).

*9) In the Introduction the authors commented on XFELs: the large sample requirement is a limitation of current delivery methods and data analysis (i.e., without post-refinement), not due to XFELs per se. However, the comment on the limited availability of beam time is well taken. In this respect, the low-dose method presented by the authors could be an important staging method for XFEL experiments, and the combination of both experiments could be very powerful*.

We added the phrase “the current implementation of this technology” to differentiate that this is, as stated, a limitation of the current delivery methods, and not of the technique itself.